# Real-Time 3D Reconstruction Method Based on Monocular Vision

**DOI:** 10.3390/s21175909

**Published:** 2021-09-02

**Authors:** Qingyu Jia, Liang Chang, Baohua Qiang, Shihao Zhang, Wu Xie, Xianyi Yang, Yangchang Sun, Minghao Yang

**Affiliations:** 1Guangxi Key Laboratory of Image and Graphics Intelligent Processing, Guilin University of Electronic Technology, Guilin 541004, China; jqy_kin@163.com (Q.J.); changl@guet.edu.cn (L.C.); shihao_zhang@yeah.net (S.Z.); xiewu588@guet.edu.cn (W.X.); 2Research Center for Brain-Inspired Intelligence (BII), Institute of Automation, Chinese Academy of Sciences (CASIA), Beijing 100190, China; sunyangchang2020@ia.ac.cn (Y.S.); mhyang@nlpr.ia.ac.cn (M.Y.)

**Keywords:** monocular vision, YOLACT++, deep optimization, real-time 3D reconstruction

## Abstract

Real-time 3D reconstruction is one of the current popular research directions of computer vision, and it has become the core technology in the fields of virtual reality, industrialized automatic systems, and mobile robot path planning. Currently, there are three main problems in the real-time 3D reconstruction field. Firstly, it is expensive. It requires more varied sensors, so it is less convenient. Secondly, the reconstruction speed is slow, and the 3D model cannot be established accurately in real time. Thirdly, the reconstruction error is large, which cannot meet the requirements of scenes with accuracy. For this reason, we propose a real-time 3D reconstruction method based on monocular vision in this paper. Firstly, a single RGB-D camera is used to collect visual information in real time, and the YOLACT++ network is used to identify and segment the visual information to extract part of the important visual information. Secondly, we combine the three stages of depth recovery, depth optimization, and deep fusion to propose a three-dimensional position estimation method based on deep learning for joint coding of visual information. It can reduce the depth error caused by the depth measurement process, and the accurate 3D point values of the segmented image can be obtained directly. Finally, we propose a method based on the limited outlier adjustment of the cluster center distance to optimize the three-dimensional point values obtained above. It improves the real-time reconstruction accuracy and obtains the three-dimensional model of the object in real time. Experimental results show that this method only needs a single RGB-D camera, which is not only low cost and convenient to use, but also significantly improves the speed and accuracy of 3D reconstruction.

## 1. Introduction

Real-time 3D reconstruction technology is not only a scientific problem that has been widely studied, but also the core technology of many applications, including robotics, virtual reality, digital media, human–computer interaction, intangible cultural heritage protection, etc.

Traditional real-time 3D reconstruction methods mainly rely on ordinary RGB cameras to collect images. Dalit Caspi proposed a method that can adapt the projection mode and number according to the characteristics of the scene. It can reduce the number of projection patterns to the necessary limit to achieve the accuracy and robustness of the Gray code technology. It uses composite structured light to complete reconstruction. In addition, Thomas P. Koninckx et al. proposed an adaptive structured light method [1], which combines geometric coding and color coding to balance reconstruction speed and quality through weights. M. Takeda first proposed the Fourier Contour Transformation (FTP) method to obtain the depth information of an object. On this basis, the literature completed the three-dimensional real-time reconstruction of the surface of complex objects [2,3]. However, such methods not only require complex and expensive hardware facilities, which lack convenience, but also have low reconstruction accuracy and poor realism of the three-dimensional model.

In recent years, with the emergence of different types of depth camera sensors, a more convenient and economical research method has been provided for the research of 3D reconstruction. More and more real-time 3D reconstruction technologies based on depth cameras have been continuously proposed [2,3,4,5,6,7,8,9,10]. Henry et al. adopted a RGB-D closest point iteration method that makes full use of color information and depth information [11], and proposed a framework called RGB-D Mapping. In this framework, they studied how to use RGB-D cameras to build dense 3D maps of indoor environments. By combining vision and depth information for view-based closed-loop detection, pose optimization is performed to achieve a globally consistent map. On this basis, Izadi et al. designed an interactive 3D reconstruction system KinectFusion, which allows users to hold the Kinect camera to move, and only use the depth information collected by the Kinect camera to track the 3D posture of the sensor. It uses the depth data obtained by the sensor in real time to create accurate geometrical 3D models. Vogiatzis et al. modeled the real-time 3D reconstruction problem from a probabilistic perspective [12], and proposed a novel video based MVS algorithm, which is suitable for real-time interactive 3D modeling. The key idea is the per-pixel probability depth estimation scheme, which updates the posterior depth distribution in each new frame and introduces the idea of SLAM (Simultaneous Localization and Mapping) to solve the real-time 3D reconstruction problem [13,14]. Furukawa et al. proposed a new multi-view stereo vision algorithm for 3D reconstruction [15]. The algorithm outputs a set of dense small rectangular blocks which is covering the visible surface in the image. The key to the performance of the proposed algorithm is an effective technique that enforces local luminosity consistency and global visibility constraints. It is also a concise effective method to convert the generated patch model into a grid, which can be further refined by an algorithm that enforces photometric consistency and regularization constraints. It automatically detects and discards outliers and obstacles, and does not require any initialization in the form of visual hulls, bounding boxes, or effective depth ranges. Many researchers have improved this method on this basis [16]. The above method is very computationally intensive and can only be run in an offline mode. However, its theory has great reference value for real-time 3D reconstruction research. Newcombe et al. proposed a real-time 3D reconstruction system based on multi-view constraints [17]. It is a system for real-time camera tracking and reconstruction. It does not rely on feature extraction, but on dense per-pixel methods. When a single handheld RGB camera flies over a static scene, a detailed texture depth map of selected key frames is estimated to generate a patchwork of surfaces with millions of vertices. Aiming at the 3D structure restoration process in real-time 3D reconstruction problems, they proposed a Cost Volume-based 3D structure Recovery method [18,19]. Yang et al. developed a real-time 3D reconstruction system suitable for UAVs on Jetson TX2 [20]. Jetson TX2 is an embedded computer developed by NVIDIA. This system uses a GPU-accelerated semi-global method to reconstruct the three-dimensional scene taken by the drone. In addition, Schöps et al. developed a real-time 3D reconstruction system on the Jetson TX1 embedded computer [21]. This system achieves a depth estimation method based on multi-resolution, by considering two different perspectives of the original image and after down sampling. The constrained relationship between image pixels reconstructs a three-dimensional model of the scene. In addition, the silhouette-based method is often used for real-time 3D reconstruction [22,23]. It can extract the contour of the foreground object in each image and reconstruct the 3D model. Although their time complexity is low, they cannot reconstruct details, especially for concave surfaces. We need to extract the contours of objects robustly. Tong et al. proposed an efficient system that uses three Kinects and one rotating stage to scan and generate models offline [24]. First, a rough mesh template is constructed and used to deform successive frames pairwisely. Second, global alignment is performed to distribute errors in the deformation space, which can solve the loop closure problem efficiently [25]. On this basis, Maimone et al. proposed an efficient system with six Kinects [26], which combines the individual three-dimensional grids from each Kinect only in the rendering stage to produce an intermediate stereoscopic view. It presents a high quality effect. The 3D reconstruction method proposed by Alexiadis et al. uses multiple Kinects to generate a single 3D mesh model in real time [27]. In order to further solve the problem of non-smooth surface and holes in the reconstructed 3D mesh model [28], Alexiadis et al. proposed a volumetric method, which implements real-time 3D reconstruction by parallelizing it on a graphics processing unit [29].

Although the above methods are mainly aimed at real-time 3D reconstruction, they still have the following shortcomings:There are many cameras and various sensors required for reconstruction, which are expensive and poor in portability.The reconstruction speed is slow. The reconstruction method is computationally expensive and time-consuming. It cannot meet real-time requirements.The reconstruction error is large, especially for the depth error. The reconstruction model effect is poor.

In order to solve the above problems, in this research, we only use a single depth camera and propose a real-time 3D reconstruction method, which can quickly and accurately obtain a 3D model of the scene. The contributions of this research are as follows:We use a single depth camera for real-time collection of visual information, and use the powerful real-time segmentation capabilities of the YOLACT++ network to extract the real-time collected information [30,31,32]. Only part of the information of the extracted items is reconstructed to ensure the real-time performance of the method.We propose a visual information joint coding three-dimensional restoration method (VJTR) based on deep learning [33,34]. This method combines three stages of deep recovery, deep optimization, and deep fusion. Taking advantage of the high accuracy and fast running speed of ResNet-152 network, through joint coding of different types of visual information, the three-dimensional point cloud with optimized depth value corresponding to the two-dimensional image is output in real time [35,36,37,38,39,40]. At the same time, the most critical visual information combination in the process of this method is determined, so as to ensure the accuracy of the reconstruction of the method.For the outliers generated in the process of reconstructing the scene, we propose an outlier adjustment method based on cluster center distance constrained (BCC-Drop) to ensure the reconstruction of the space of each object consistency and reconstruction accuracy.We propose a framework organization method that can use a single depth camera to quickly and accurately perform 3D reconstruction. It is without any human assistance or calibration, and can automatically organize the reconstructed objects in the 3D space.Experimental results show that our method greatly improves the performance of 3D reconstruction and is always better than other mainstream comparison methods. The reconstruction speed reaches real-time and can be used for real-time reconstruction.

The rest of the paper is organized as follows: the proposed method is explained in Section 2, the experiment and results are shown in Section 3, and the conclusions are drawn in Section 4.

## 2. Materials and Methods

### 2.1. Framework

Our proposed method is shown in Figure 1. The framework consists of three steps. Firstly, a single RGB-D camera is used to collect visual information, and the YOLACT++ network is used to segment the input image to extract the visual information that needs to be reconstructed. Secondly, for the segmented visual information, the VJTR method is used to obtain the three-dimensional point information of the object. The detailed network structure diagram of the YOLACT++ and VJTR will be given as Figure 1 and Figure 2 in the following chapters. Finally, we propose an outlier adjustment method based on cluster center distance constrained (BCC-Drop) to remove the outliers in the reconstruction process, which is used to reduce the object reconstruction error and improve the 3D reconstruction accuracy. We save each 3D point generated through the VJTR network as a one-dimensional array, obtain the color information of the point from the RGB image according to the two-dimensional pixel value of the point, and store the color information of the 3D point and the 3D point in the same array. The 3D point set is visualized by OpenGL, and the fuzzy reconstruction image is obtained. By processing the 3D point set generated by the VJTR network with the BCC-Drop method, accurate 3D point set information is obtained. Then, the accurate three-dimensional point set information is reconstructed by the above method to obtain a clear reconstruction image. Experimental results show that this method only needs a single RGB-D camera, which is not only low cost and convenient to use, but also significantly improves the speed and accuracy of 3D reconstruction.

### 2.2. Visual Information Segmentation and Extraction

We use the YOLACT++ network to segment the visual information collected by the RGB-D camera in real time. As shown in Figure 2, the RGB image collected in real time is passed to the YOLACT++ network, which is used for information extraction, to obtain the segmented image RGB’.

The input image size of the YOLACT++ model is 550×550 and the Backbone used is ResNet101. Five convolution modules of the ResNet101 are used in YOLACT++. They are conv1, conv2_x to conv5_x, corresponding to C1, C2 to C5 in Figure 2. For each standard 3×3 convolutional layer in C3–C5, in order to achieve trade-off in terms of time and performance, we replace it with variability convolution every three convolutional layers [41].

P1 to P5 is the FPN network in Figure 2. It obtains P3 from C5 first through a convolutional layer, then uses bilinear interpolation on P3 to double the feature map, and adds it to the convolutional C4 to obtain P2. Then, we used the same method to find P1. We convolved and down sampled P3 to find P4, and performed the same convolution and down sampling on P4 to find P5, thereby establishing an FPN network. The advantage of using FPN is that it can enrich the features learned by the model. The next step is the parallel operation. P1 is sent to Protonet, and P1–P5 are also sent to Prediction Head at the same time. The Protonet network module is composed of several convolutional layers. Its input is P1, and its output mask dimension is 138×138×k(k=32), that is, 32 prototype masks with the size of 138×138. The branch of Prediction Head is improved on the basis of RetinaNet [42]. It uses a shared convolutional network which can increase the speed and achieve the purpose of real-time segmentation. Its input is a total of five feature maps P1–P5. Each feature map generates anchors first. Each pixel generates 3 anchors with ratios of 1:1, 1:2, and 2:1. The anchor basic side lengths of the five feature maps are 24, 48, 96, 192, and 384, respectively. The basic side length is adjusted according to different proportions to ensure that the area of the anchor is equal.

The mask coefficient obtained by the prediction head and the prototype mask obtained by the protonet are subjected to matrix multiplication to obtain the mask of each target object in the image. In the second half of the model, a mask re-scoring branch was added. This branch uses the cropped prototype mask (non-threshold) generated by YOLACT as input, and outputs the IoU corresponding to each category of the GT mask. The fast mask re-scoring branch consists of 6 convolutional layers and 1 global mean pooling layer. Crop refers to clearing the mask outside the boundary. The boundary of the training phase is the ground truth bounding box, and the boundary of the evaluation phase is the predicted bounding box. Threshold refers to the image binarization of the generated mask with a threshold of 0.5.

### 2.3. Visual Information Joint Coding Three-Dimensional Restoration Method

After we obtain the segmented visual information, as the segmentation is performed along the contour of the object edge, the depth value of the contour edge is prone to large errors at this time. We first introduce the mathematical representation of the process of this method from the RGB image and the Depth image, respectively. Then, we introduce how to use the convolutional neural network (CNN) to obtain the estimated value of the three-dimensional position from the visual information of a single RGB-D camera. We call 3D point reconstruction using a convolutional neural network as VJTR.

#### 2.3.1. Reconstruction of 3D Coordinates from RGB Image and Depth Image

Assuming that a certain pixel in a two-dimensional RGB image is represented by $I_{I}={\left[u,v,1\right]}^{T}$, and its corresponding three-dimensional reconstruction point is represented by $W_{I}={\left[X_{I},Y_{I},Z_{I}\right]}^{T}$, the process of mapping RGB image pixels to three-dimensional reconstruction positions is described as
(1)$$\tau I_{I}=F_{I}W_{I},\;\mathrm{where}\;F_{I}=\left[\begin{array}{ccc} f_{x} & \eta & u_{0}\\ 0 & f_{y} & v_{0}\\ 0 & 0 & 1 \end{array}\right].$$

In Equation (1), τ is an arbitrary scale factor. $F_{I}$ is the intrinsic matrix of the camera, $u_{0}$ and $v_{0}$ are the coordinates in the image coordinate system. $f_{x}$ and $f_{y}$ are the image scale factors along the u and v axes, respectively, and η is the skew parameter of the image between u and v.

Equation (1) can be further simplified as
(2)$$I_{I}=M_{I}W_{I},\;\mathrm{where}\;M_{I}=\frac{1}{\tau }F_{I}.$$

Setting the U, V value of pixel I of RGB image and the camera intrinsic matrix to $F_{I}$, we can use Equation (2) to calculate the $X_{I}$, $Y_{I}$ mapped to the three-dimensional space from a single RGB image. However, for $Z_{I}$, the true distance from the camera is still uncertain.

Similarly, the relationship between 3D reconstruction point $W_{D}={\left[X_{D},Y_{D},Z_{D}\right]}^{T}$ and Depth image projection $I_{D}={\left[\alpha ,\beta ,\gamma \right]}^{T}$ is
(3)$$\tau I_{D}=F_{D}W_{D},\;\mathrm{where}\;F_{D}=\left[\begin{array}{ccc} d_{11} & d_{12} & d_{13}\\ d_{21} & d_{22} & d_{23}\\ d_{31} & d_{32} & d_{33} \end{array}\right].$$

τ is an arbitrary scale factor, $F_{D}$ is the sensor parameter of the Depth image given by the 3×3 matrix, which is different from the RGB imaging pinhole model and cannot be directly calculated by the geometric method.

Equation (3) can be further simplified as
(4)$$I_{D}=M_{D}W_{D},\;\mathrm{where}\;M_{D}=\frac{1}{\tau }F_{D}.$$

Combining Equations (2) and (4) together, we can obtain
(5)$$\left[\begin{array}{c} I_{I}\\ I_{D} \end{array}\right]=\left[\begin{array}{cc} M_{I} & 0\\ 0 & M_{D} \end{array}\right]\left[\begin{array}{c} W_{I}\\ W_{D} \end{array}\right].$$

That is, the three-dimensional position of the pixel can be estimated synchronously from its RGB and Depth images.

Equation (5) can be simply written as
(6)I=MW,
where $I={\left[I_{I},I_{D}\right]}^{T}$, $W={\left[W_{I},W_{D}\right]}^{T}$. The M is a diagonal matrix constructed by $M_{I}$ and $M_{D}$, W could be obtained from $M^{-1}I$.

#### 2.3.2. Simultaneous Estimation of Three-Dimensional Values Using ResNet-152 Network

High-dimensional image data can be presented by low-dimensional code using deep architecture or CNN-based autoencoder. Similarly, an autoencoder can be used to represent a two-dimensional point from the position of the white pixel embedded in the black background image. In this article, we refer to this image as a two-dimensional position mask (2DM) image. The second row of image is samples of some 2DM images in Figure 3. In this way, the CNN network provides a solution to reconstruct the three-dimensional position of points from the two-dimensional description of RGB images, grayscale Depth images, and 2DM images.

In Equation (6), the symbol I contains the 2D point in RGB image and D in depth image. Supposing an auto-encoder operator A transfer I to P with P=AI, where the number of A’s columns equal to that of I’s rows. Accordingly, while Equation (6) is left multiplied by A, it is transferred to
(7)P=AI=AMW,
where P is the coding matrix of RGB image, Depth image, and 2DM image for I. From Equation (7), we can find
(8)W=KP,
where the unknown matrix K can be obtained by $K={\left(AM\right)}^{-1}$. In Equation (8), W, P are the joint coding of the three-dimensional point and its two-dimensional point obtained by the auto-encoder. When the RGB-D camera is turned on, the camera can automatically generate multiple joint codes P from the RGB, Depth image, and the corresponding 3D position W automatically. The VJTR method proposed in this paper is to obtain K from each of the variables P and W. Our objective function is expressed as
(9)$$K=\arg_{\forall (P,W)}\min\left\|KP-W\right\|.$$

From Equation (9), we can get the estimated value of the true three-dimensional position of the $K_{real}$ in the world coordinate system.

As shown in Figure 3, we use the ResNet-152 network to simultaneously estimate the three-dimensional position. We will jointly encode the RGB’ image and 2DM image, which are segmented by the YOLACT++ network, and the Depth image collected by the RGB-D camera into the ResNet-152 network, and finally output the 3D estimated position of the object. The ResNet-152 network is similar to other residual networks. The convolution module has 5 modules from conv1, conv2_x to conv5_x. In conv1, we use the convolution kernel of 7×7 to convolve the input image of 224×224×3. This layer has 64 channels with a step size of 2 to obtain a 112×112×64 feature map. Then, we input the feature map to the maximum pooling layer with a step size of 2, and a convolution kernel of 3×3 to perform feature extraction. Finally, we obtain a 56×56×64 feature map, and input it into the residual layer. The specific structure of the residual layer is shown in Table 1.

### 2.4. Reconstruction Error Correction

In the process of obtaining 3D reconstruction point set P which uses the above method, some outliers will be generated. Therefore, we propose an outlier adjustment method based on cluster center distance constrained (BCC-Drop) to improve the reconstruction accuracy.

Assume that P contains m points: $p_{1},\ldots p_{i},\ldots p_{m}$, where $p_{i}\in R^{S}$, its initial cluster center point $p_{0}$ is derived from $p_{0}=\sum_{i=1}^{m}p_{i}/m$. For each point $p_{i}$, we set a corresponding new set of N, the set $n_{i}=\left\|p_{i}-p_{0}\right\|(1\leq i\leq m)$ is composed of $n_{i}$. Then, according to the literature [43], the local density and distance $\delta_{i}$ for each $n_{i}$ are given as:(10)$$\rho_{i}=\sum_{j=1,j\neq i}^{m}N(d_{ij}-d_{c}),$$
(11)$$\delta_{i}=\min_{j:\rho_{j}>\rho_{i}}(d_{ij}).$$

In Equation (10), $d_{ij}$ is the distance between the terms $n_{i}$ and $n_{j}$. If n<0, the function N(n)=1, otherwise N(n)=0. $d_{c}$ is a cutoff distance, which is suggested to be the values that ensure the average number of neighbors is around 1 to 2% of the total number of points in the dataset [44]. In Equation (11), $\delta_{i}$ is measured by the minimum distance between the item $x_{i}$ and any other item with higher density. Then, according to the definition of BCC-DROP [43], the items which have the max values of δ are the cluster center points of good items, and the items with low values of ρ are outliers. We divide different points into the same cluster by $d_{c}$, and then determine whether the point is a cluster center point or an outlier by calculating the distance from the point with higher density. After finding the better set $N_{in}=N-N_{out}$, where $N_{out}$ is the abnormal item in N, we can obtain the corresponding good point set $P_{in}$. After finding $P_{in}$, we use
(12)$${\bar{x}}_{i}=(x_{i}-x_{\min})/(x_{\max}-x_{\min}),$$
(13)$${\bar{y}}_{i}=(y_{i}-y_{\min})/(y_{\max}-y_{\min}),$$
(14)$${\bar{z}}_{i}=(z_{i}-z_{\min})/(z_{\max}-z_{\min}),$$
to adjust each point of $p_{i}=(x_{i},y_{i},z_{i})\in P_{in}$. In Equations (12)–(14), the range of $x_{i},y_{i},z_{i}$ is (0, 1], where $x_{\max}=\forall_{i}\max(x_{i})$, $x_{\min}=\forall_{i}\min(x_{i})$, $y_{\max}=\forall_{i}\max(y_{i})$, $y_{\min}=\forall_{i}\min(y_{i})$, $z_{\max}=\forall_{i}\max(z_{i})$, and $z_{\min}=\forall_{i}\min(z_{i})$ are the coordinates of the maximum and minimum values of P. Finally, we obtain the point set $\bar{P}$ $\left({\bar{p}}_{i}=({\bar{x}}_{i},{\bar{y}}_{i},{\bar{z}}_{i})\in \bar{P}\right)$ which is normalized. The point set $\bar{P}$ is the real-time 3D reconstruction model of the object.

### 2.5. Method Summary

The whole process of the algorithm proposed in this article is summarized in Algorithm 1. Firstly, we adjust the image size and input them into the neural network in batches in the training phase of ResNet-152. Secondly, we use the gradient function to minimize the loss function to train the neural network. Finally, the model with the highest target detection accuracy is selected according to the validation set. In the same way, we used a similar method in the training stage of the YOLACT++ neural network.

**Algorithm 1.** The process of the real-time 3D reconstruction method1:The RGB and Depth images of the scene are collected by the monocular camera.2:Input RGB to YOLACT++ to obtain the segmented RGB’, 2DM.3:Input RGB’, 2DM, Depth to VJTR to get P.4:Set $p_{0}\leftarrow \sum_{i=1}^{m}p_{i}/m$, **then** let $\delta_{\max}\leftarrow -1.0$ and s←−1.5:**for** Each $p_{i}$6: Use Formula (10) to calculate the $\rho_{i}$.7: **for** *i* = 1 to m step **do**8:   Use Formula (11) to calculate the $\delta_{i}$.9:   **if** ($\delta_{i}>\delta_{\max}$) **then**10:     $\delta_{\max}\leftarrow \delta_{i}$. 11:     s←i.12:   **end if**13: **end for**14:
**end for**
15:**for** *i* = 1 to m step **do**16: **if**$\left(\left\|p_{i}-p_{0}\right\|<\lambda \left\|p_{s}-p_{0}\right\|\right)$
**then**17:   Add $p_{i}$ to $P_{in}$.18: **else**19:   Add $p_{i}$ to $P_{out}$.20: **end if**21:
**end for**
22:**for** Each $p_{i}$23:  Use Formulas (12)–(14) to normalize its coordinate values.24:  Add $p_{i}$ to $\bar{P}$.25:
**end for**
26:**return** $\bar{P}$.

## 3. Experiments

In this section, we first introduce the experimental environment and visual information extraction results, and compare the segmentation performance of different instance segmentation methods on the COCO dataset. Then, we describe the implementation of VJTR and compare the experimental results of several different types of input combinations and different types of CNN 3D position estimation. After that, we compare the model obtained by using the neural network for 3D position estimation with the model obtained by using the BCC-Drop method. NN represents the experimental results which use the neural network. BCC-Drop-NN represents the experimental results which use the neural network first to obtain the three-dimensional scene point cloud, and then use the BCC-Drop method to remove outliers. Finally, we detail the time cost of each step for the proposed method. In the experimental results section, we add a comparison between our method and other main methods on the YCB-Video dataset.

### 3.1. Experimental Setting

This experiment is based on Windows 64-bit platform. We use the development language C++ as it has the advantages of high efficiency, simplicity, complete third-party libraries, and portability. The third-party development libraries used are OpenCV and OpenGL. OpenCV is used to process image data, and OpenGL is used for real-time reconstruction and visualization. The handheld RGB-D camera dataset is collected by the kinect2.0 camera. The RGB-D camera is connected to a 3.2 GHz i7-8700 CPU, 16.0 G RAM, and NVidia GTX 1660 graphics computer. We use the standard dataset COCO to train, test and validate the YOCACT++ model. For the VJTR method, we use 11,356 sets of data collected by a handheld RGB-D camera as the training set, and use the standard data-set YCB-Video to test and verify the effectiveness of our method.

### 3.2. Implementation and Results of Visual Information Segmentation Extraction

We use YOLACT++ network to extract objects from RGB images. Before inputting the images to the YOLACT++ network, we adjust them to images with a resolution of 550×550. We use different types of instance segmentation networks to segment and extract visual information on the COCO dataset. As shown in Table 2, the Mask Scoring R-CNN network has the best segmentation effect, but its average running time is as high as 116.3 ms. The extraction of visual information will greatly reduce the speed of real-time reconstruction in this way. Compared with other mainstream segmentation networks, the YOLACT++ network can reach 33.5 FPS, which meets the needs of real-time reconstruction. The average precision is 34.1. It can be seen that under the premise of ensuring real-time segmentation, YOLACT++ can segment objects more accurately and reduce reconstruction errors.

### 3.3. VJTR Method Realization and Results

We input the image RGB’, 2DM, and Depth which are segmented by YOLACT++ to different types of CNN networks. The experimental results of deep recovery are listed in Table 3, in which we select VGG-16, InceptionNet-V3 and ResNet-152 as different types of input combinations. In the input column of Table 3, C, D, and M, respectively represent RGB’ image, Depth, and 2DM mask image. As the input requires the depth map corresponding to the RGB image, we use the data collected by our handheld RGB-D camera as the training set, and use the public dataset YCB-Video for verification.

From Table 3, we can see that ResNet network used as dimension reduction for C, D, M performs better than those of VGG and InceptionNet on *AE* and *ME*. We believe that the smaller the average error and maximum error, the better the performance. We can find that D+M (2DM and Depth) is the basic effective channel used for depth recovery, while the channel C (RGB’ image) has little help in reducing the depth recovery error. It only provides two-dimensional image information that requires three-dimensional reconstruction. Secondly, we can see that the maximum reconstruction error value of D is much greater than the maximum reconstruction error value of D+M. Due to the measurement error caused by backlight, objects outside the reflection range, or distant objects, and multiple infrared reflections in the environment, there are a lot of noise and pits in the Depth. A single pixel may be located in the noise range or recessed, resulting in a lot of inaccurate depth information. M (2DM mask image) can provide 5×5 average information which can reduce the probability of maximum errors in visual reconstruction points. As shown in Figure 4, Figure 4a is the distance of the object’s red line relative to the real ground, and Figure 4b is the point cloud model obtained by deep fusion of the RGB image collected by RGB-D and the Depth image, Figure 4c is the model obtained by the VJTR method. We use OpenGL to visualize the point cloud information generated in this article, and at the same time obtain the color information of the point in the RGB image through the pixel value. For the teacup handle, due to its irregular shape and small volume, the depth value is very easy to take an abnormal value or a null value. We can clearly see that the proposed VJTR method effectively reduces the depth error in the reconstruction of the contour edge of the object. In addition, we can see that it is time-consuming to restore the depth values of all points of the object from the image, so we only reconstruct the selected k points in the actual process. Suppose the number of object points obtained through the YOLACT++ network is w, where k=w/5.

### 3.4. BCC-Drop Strategy Implementation and Results

Figure 5b,c show the segmentation results and selected partial contour points of m objects obtained by YOLACT++. The w and k values of bottle, blue barrel, brown box, blue cans, and red bowl from top to bottom are 684, 440, 2680, 260, and 468, and 137, 88, 536, 52, and 94, respectively. When we determine the points that need to be reconstructed, we use ResNet-152 neural network to estimate them.

Figure 5d,e show the object reconstruction obtained by a single neural network (NN) method and the BCC-Drop strategy proposed for Figure 5a, where the red line in Figure 5b represents the true ground distance between two contour points. The corresponding red lines in Figure 5d,e indicate that the reconstruction distance of NN and BCC-Drop-NN, respectively. Assuming that the length of the red line ground true distance is $L_{j}^{GT}$, the red line reconstruction length NN and BCC-Drop-NN are, respectively $L_{j}^{NN}$ and $L_{j}^{DP}$ where j(0≤j≤J) is the total number of red line segments for different positions of the object, to be used to compare experimental results. We use
(15)$$Err_{NN}=\frac{\sum_{j=1}^{J}e_{j}^{NN}}{J},\;\mathrm{where}\;err_{NN}=\frac{\left\|L_{j}^{NN}-L_{j}^{GN}\right\|}{L_{j}^{GN}},$$
(16)$$Err_{DP}=\frac{\sum_{j=1}^{J}e_{j}^{DP}}{J},\;\mathrm{where}\;err_{DP}=\frac{\left\|L_{j}^{DP}-L_{j}^{GN}\right\|}{L_{j}^{GN}},$$
to calculate the reconstruction error of NN and BCC-Drop-NN, respectively.

The values in the accuracy column of Table 4 represent the absolute and relative errors of the object given in Figure 5a, where $Err_{NN}$ and $Err_{DP}$ the relative errors. For each object, we calculate at least twelve lines and their distances, that is J≥12. It can be seen from Table 4 that the absolute error of reconstruction obtained by BCC-Drop-NN is significantly smaller than that of NN reconstruction. Although similar reconstruction effects are obtained by the BCC-Drop-NN method and NN, we can clearly see that the relative error of the bottle is about 9.16% and 12.29%, respectively, and the blue barrel is about 5.65% and 9.71%, respectively or the red bowl, its values are about 6.53% and 10.75%, respectively. From this experimental result, the relative error obtained by our proposed BCC-Drop-NN method is smaller than that of the NN method. In general, the average total error value of the BCC-Drop-NN method is about 3.7% less than the average total error value of the NN method.

The three columns of time cost in Table 4 detail the time cost of each stage. It is worth noting that for a given RGB image, the time overhead of YOLACT++ is about 30.11 ms, which has nothing to do with the number of objects in the scene. This is as YOLACT++ always calculates the probability of whether each pixel in the image is a boundary point of an object. For the depth estimation and anomaly elimination steps, the average cost time of these two steps is about 2.32 ms and 0.90 ms. The time cost of these two parts is closely related to the number of scene reconstruction points, so we only select some points for reconstruction. The reconstruction time cost of this method is about 0.033 s, 30 or 31 frames per second (FPS).

### 3.5. Experimental Results

Figure 6 shows the reconstruction effects of different methods on the same public dataset YCB-Video. We can see that, although the method in Figure 6b reconstructs most of the elements of the scene, many of these elements need not to be reconstructed. This method not only wastes a lot of time, but also has low reconstruction accuracy. Although the method in Figure 6c has a fast reconstruction speed, its reconstruction depth value error is relatively large, which makes the object broken after reconstruction. It fails to complete the reconstruction. In addition, in the scene in Figure 7a, we can see that these objects are placed closely together, there is no distance between them, and there is obvious overlap and occlusion. For example, a blue box is placed on a white box, and the white box is blocked by a dark cup and a white cup. Even so, it can be seen from Figure 7d–f that the reconstruction points of these objects are correctly positioned at their respective positions. Take the white box as an example. In the views shown in Figure 7a,b, a white cup and a dark gray cup are placed to block, and there is even a blue box stacked with it. Two objects can still be distinguished well in the reconstruction view of Figure 7d,f. It shows that, even if objects are placed closely or even stacked in the RGB-D image, their 3D reconstruction points are distributed correctly in space, and their spatial relationship in the real 3D space is also well reflected.

In addition, it can be seen from the reconstruction results in Table 4 and the real 3D reconstruction results in Figure 7 that the proposed method can complete real-time 3D reconstruction of monocular vision. With an average absolute reconstruction error of 5.37 mm and a relative error of 6.45%, this method can achieve accurate reconstruction of objects in a virtual three-dimensional space, even if these objects are closely placed in the scene in a stacked and overlapping manner. It lists the comparison of time cost and accuracy between our real-time 3D reconstruction method and other real-time 3D reconstruction methods in Table 5. It is not difficult to see that the real-time reconstruction method based on monocular vision proposed by us not only requires a single camera to complete real-time reconstruction, but also greatly improves the reconstruction speed and reconstruction accuracy.

## 4. Conclusions

In this work, we propose a real-time reconstruction method based on a monocular camera. The YOLACT++ network is used to extract the objects that need to be reconstructed in the RGB-D image from the clustering environment. We use ResNet-152 neural network to propose a three-dimensional position estimation method (VJTR) for joint coding of visual information. This method combines three stages of depth recovery, depth optimization, and depth fusion to quickly and accurately obtain the three-dimensional point value of the object extracted by the YOLACT++ network from the RGB-D image. A method is proposed to limit outlier adjustment based on the distance of cluster centers, namely the BCC-Drop method. It can remove the partial separation group value and reduce the three-dimensional reconstruction error. We verified the effectiveness of this method on the actual complex stacked 3D scene and the public dataset YCB-Video. The experimental results show that the reconstruction speed of this method is 30 FPS, and the relative error of reconstruction is 9%. Compared with other mainstream real-time 3D reconstruction methods, this method has stronger real-time performance on ordinary PCs and better real-time reconstruction effects in a real cluster environment. In addition, our method can automatically organize the reconstructed objects in the three-dimensional space, and extract and label the reconstructed objects through the YOLACT++ network in a clustering environment. Using these tags obtained by the YOLACT++ network, the reconstructed points of the object can also be automatically and synchronously marked in the three-dimensional space. It should be noted that our method does not optimize the phase of RGB-D camera acquisition of two-dimensional images, which will be our future work.

## Figures and Tables

**Figure 1 sensors-21-05909-f001:**
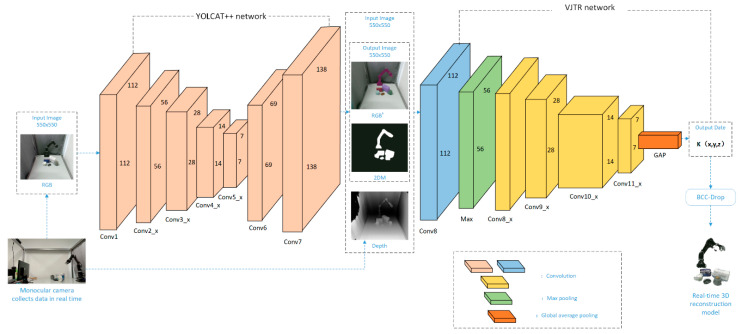
Flow chart of the method.

**Figure 2 sensors-21-05909-f002:**
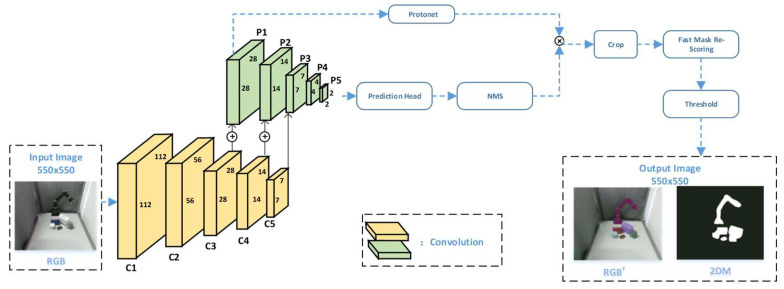
The network structure of YOLACT++.

**Figure 3 sensors-21-05909-f003:**
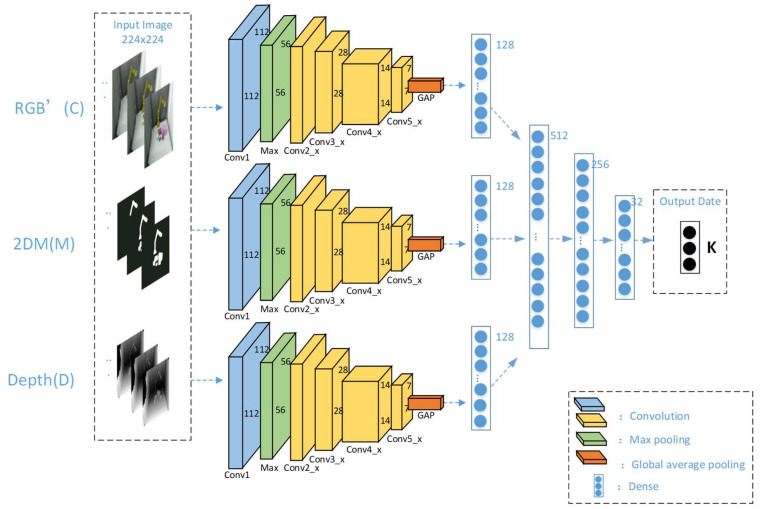
Schematic diagram of VJTR method.

**Figure 4 sensors-21-05909-f004:**
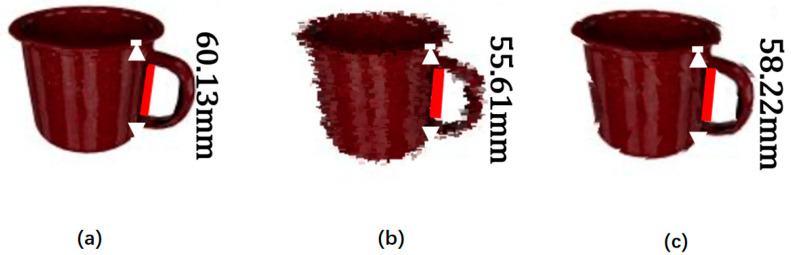
Comparison of teacup reconstruction models on YCB-Video dataset. (**a**) RGB image; (**b**) NN results; and (**c**) BCC-Drop results.

**Figure 5 sensors-21-05909-f005:**
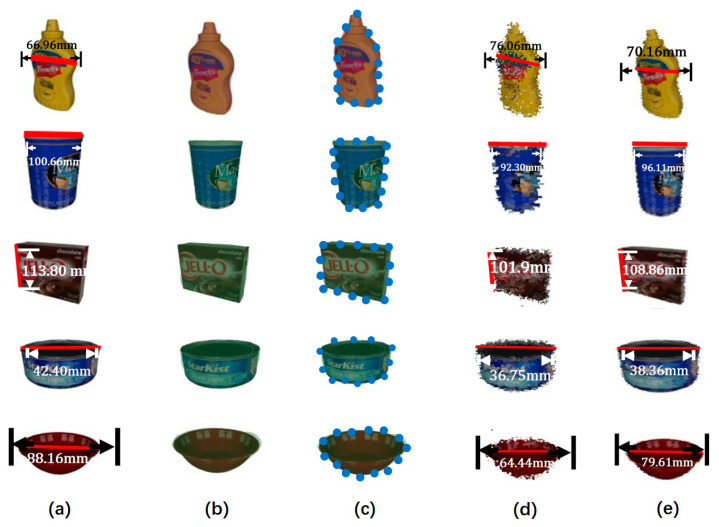
Extract and reconstruct samples of some objects in the experiment. (**a**) RGB image; (**b**) YOLACT++ segmentation results; (**c**) Schematic diagram of selected contour points; (**d**) NN result; and (**e**) BCC-Drop results.

**Figure 6 sensors-21-05909-f006:**
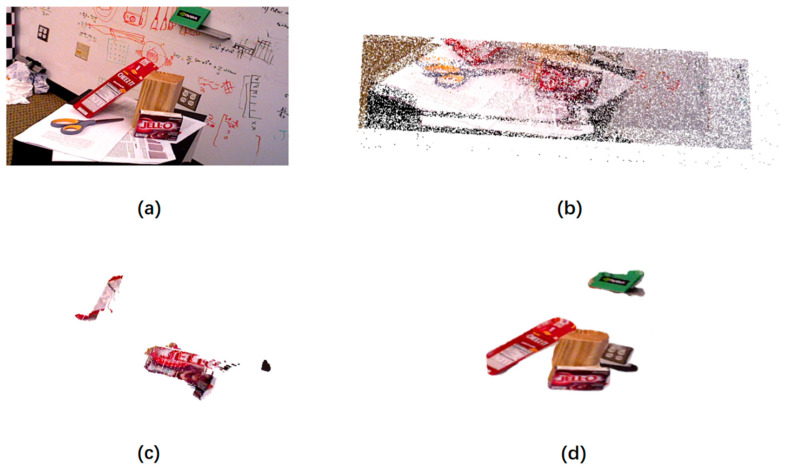
The experimental results of our method and other main methods on the YCB-Video dataset. (**a**) YCB-Video data RGB image; (**b**) ORB reconstruction results; (**c**) EKF-SLAM reconstruction results; and (**d**) OURS reconstruction results.

**Figure 7 sensors-21-05909-f007:**
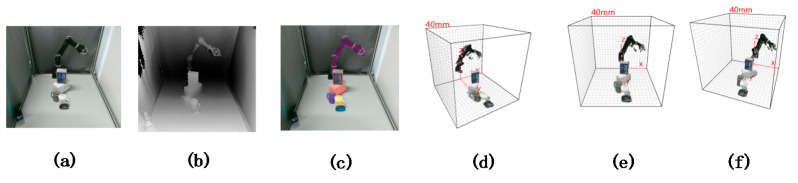
Real-time reconstruction results of monocular cameras in complex scenes. (**a**) RGB image; (**b**) Depth image; (**c**) Object extraction results obtained by YOLACT++; and (**d**–**f**) Real-time 3D reconstruction results from different views.

**Table 1 sensors-21-05909-t001:** Detailed parameters of hourglass structure.

**Network Layer**	Kernel Parameter	Repeat	Output
Input	56×56×64		
Block1	$\left[\begin{array}{c} 1\times 1,64\\ 3\times 3,64\\ 1\times 1,256 \end{array}\right]$	3	56×56×64
Block2	$\left[\begin{array}{c} 1\times 1,128\\ 3\times 3,128\\ 1\times 1,512 \end{array}\right]$	8	28×28×128
Block3	$\left[\begin{array}{c} 1\times 1,256\\ 3\times 3,256\\ 1\times 1,1024 \end{array}\right]$	36	14×14×256
Block4	$\left[\begin{array}{c} 1\times 1,512\\ 3\times 3,512\\ 1\times 1,2048 \end{array}\right]$	3	7×7×512

**Table 2 sensors-21-05909-t002:** Segmentation experiment results of different types of instance segmentation networks on the COCO dataset.

Number	Method	Average Precision	FPS	Time (ms)
1	PA-Net [45]	36.6	4.7	212.8
2	RetinaMask [46]	34.7	6.0	166.7
3	FCIS [47]	29.5	6.6	151.5
4	Mask R-CNN [48]	35.7	8.6	116.3
5	Mask Scoring R-CNN [49]	38.3	8.6	116.3
6	YOLACT++	34.1	33.5	29.9

**Table 3 sensors-21-05909-t003:** The results of joint coding of different input combinations and different types of CNN depth recovery on the YCB-Video dataset.

Number	Neural Networks Category	Enter	Average Error (m)	Maximum Error (m)	Parameter Size (MB)	Time (s)
1	VGG-16	C + D + M	0.041	0.098	46.20	0.159
2		C + D	0.056	0.113	39.96	0.176
3		D + M	0.040	0.079	21.71	0.169
4		D	0.030	0.101	20.04	0.159
5	InceptionNet-V3	C + D + M	0.049	0.094	27.77	0.151
6		C + D	0.051	0.104	22.63	0.13
7		D + M	0.026	0.063	21.63	0.129
8		D	0.031	0.096	20.04	0.125
9	ResNet-152	C + D + M	0.037	0.085	13.47	0.121
10		C + D	0.044	0.068	12.61	0.115
11		D + M	0.017	0.052	8.79	0.115
12		D	0.041	0.070	6.52	0.096

**Table 4 sensors-21-05909-t004:** The reconstruction accuracy and time cost on the YCB-Video dataset.

Object	Point		Precision				Time (ms)		
	*m*	k	$\frac{\sum_{j}(L_{j}^{NN}-L_{j}^{GT})}{J}$	$\frac{\sum_{j}(L_{j}^{DP}-L_{j}^{GT})}{J}$	$Err_{NN}$ (%)	$Err_{DP}$ (%)	*T* _YOLACT++_	T _VJTR_	*T* _BCC-Drop_
Bottle	684	137	8.62 mm	6.43 mm	12.29	9.16	30.12	2.32	1.14
Blue barrel	440	88	9.77 mm	5.69 mm	9.71	5.65	30.01	1.35	0.44
Brown box	2680	536	11.13 mm	6.94 mm	9.78	6.10	30.33	5.26	2.21
Blue cans	260	52	3.58 mm	2.04 mm	8.44	4.81	30.06	1.22	0.23
Red bowl	468	94	9.48 mm	5.76 mm	10.75	6.53	30.03	1.48	0.51
Mean	/	/	8.52 mm	5.37 mm	10.19	6.45	30.11	2.32	0.90

**Table 5 sensors-21-05909-t005:** The performance of our method and other main methods on the YCB-Video dataset.

Algorithm	Err (%)	Running Time (ms)	Detection Speed (fps)
EKF-SLAM	18.41	80.11	12
ORB	10.34	251.32	3
OURS	6.71	33.64	29

## Data Availability

This study analyzed publicly available data sets. These data can be found here: [https://rse-lab.cs.washington.edu/projects/posecnn/] (accessed on 28 August 2021).

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
