# Peer review of "Real-Time 3D Reconstruction Method Based on Monocular Vision"

_sensors, 2021, doi:10.3390/s21175909_

Round 1

Reviewer 1 Report

The theme of real-time 3D reconstruction in the field of computer vision is currently highly topical. Many researchers are concerned with this topic and describe their research results in many renowned journals. Today's effort is to apply machine vision to many different industries and thus facilitate human work associated with the control of various processes. The article detailed describes use the YOLACT++ network to segment the visual information collected by the RGB-D camera in real time which is used for information extraction to obtain the segmented image RGB-D from the clustering environment. This work is a combination of theory and experimental techniques. I highlight the nicely crafted "state of art" and “list of references” of authors and articles solving similar research issues from around the world. The article is written at a high level and is innovative in the field of computer vision. The article is in my opinion interesting and suitable for publication in this journal.

Small formal errors:

- lines 152 and 153 formal font size errors...I suggest editing before publishing and checking the font sizes, template and notation of the equations throughout the article,

- the "Conclusion" section of the article could be a bit more extensive and deal in more detail with the results that are presented in the core of the article.

Author Response

We thank the reviewer sincerely for your encouraging comments about our work. Thank for the detailed review of the paper and the insightful and valuable comments. We consider your comments carefully and revise the paper accordingly. We respond to every comment of the reviewer seriously.For more details, please refer to the attachment.

Reviewer 2 Report

Summary

The paper presents a method for reconstructing points of 3D objects in real time using a single RGB-D image as the input. These images are first processed by the YOLACT++ network for object segmentation. Then, a method called VJTR is used to reconstruct the 3D points of the segmented objects. The output is noisy, so the final step is a clustering-based method which can refine the 3D points by detecting outliers. The method is evaluated on a very controlled setting where both training and testing images are extracted from the same scene including a robotic arm and other objects.

Technical details.

  • Overall, the paper can be obscure at times because of missing details and connections. There are several points where portions of the method are explained out of context without at least providing references to external sources which can provide the missing information.
  • In terms of math, the paper is really unclear, containing a lot of notation abuses and poorly defined elements/equations.
  • The proposed method is motivated as a generic procedure for real time 3D reconstruction of scenes based on RGB-D images. However, the actual experimental setting described in Section 3.1 is pretty much limited to one specific scenario. Both training and testing RGB-D images come from the same scenario. In that sense, it is hard to imagine how would these results generalize to more open applications. At the same time, it is not surprising that existing generic methods get worst results than the proposed method on this particular setting for which it was developed.
  • The relationship between the architectures in Figure 1 and Figure 2 is not well stablished. Authors should provide a higher-level diagram that shows the interactions between these two networks. The main text does not make it clear either. While additional details might be available on the original YOLACT++ paper, this work should be self-contained by providing enough information.
  • Section 2.2. What is a “variability convolution”? Authors should provide a reference for this.
  • Section 2.2. What is “RetinaNet”, authors do not provide a reference for this.
  • Section 2.3.1. There is not \tau in equation 1.
  • Equations 1, 2 and 3. All of them include a multiplication of a 3x3 matrix by a 2x2 matrix, this is invalid. Is there any implicit matrix expansion here? It might be better to simply provide all matrix elements.
  • Other equations of matrices also seem to be either wrong or abusing mathematical notation.
  • Equation 10. \rho_{i} is used and mentioned but never quite defined. Also, authors should avoid using X as a function name. It is also unclear how or why is the cut-off distance d_{c} set as a proportion of the total number of points.
  • Section 2.4. “The item with the maximum value of \delta is a good cluster center point, while the item with a low value of \rho is an outlier”. Why? These variables are not well explained on the paper, and the notation for equation 11 is unclear. Due to these issues, this idea is not obvious at this point.
  • The name VOLACT appears for the first time in Section 3.2 and it is not defined before usage. It is also unclear what is the difference between VOLACT-400 and VOLACT-500++.
  • Equations 15 and 16 use the terms “e^{NN}” and “e^{DP}”, but define the terms “err^{NN}” and “err^{DP}” respectively. Note that “e^{x}” might be confused with the exponential function exp(x).

Language, presentation, formatting.

  • The paper contains multiple grammar errors.
    • “we only reconstruct part information”
    • “…, and the YOLACT++ network which is used to ….”. The same sentence appears in Section 2.1 without “which” and it reads better.
    • “From Table 3, we can see that performs better than in terms of average error and maximum error”.
    •  
  • Section 2.2. Different font sizes are used for numbers (e.g., “138x138”) every time they appear.

Author Response

(The authors gave the same response as above.)

Reviewer 3 Report

The central idea and the objectives are clearly stated and concluded in the conclusion section. The authors did a good job at organizing the paper.

(1)Few grammatical errors need to be taken care of.

(2) More references need to be added.

(3) Introduction should include more description of the peer work.

(4) Figures do not have consistent font size and color. some of the annotations are not legible.

Author Response

(The authors gave the same response as above.)

Round 2

Reviewer 2 Report

Summary

The paper presents a method for reconstructing points of 3D objects in real time using a single RGB-D image as the input. These images are first processed by the YOLACT++ network for object segmentation. Then, a method called VJTR is used to reconstruct the 3D points of the segmented objects. The output is noisy, so the final step is a clustering-based method which can refine the 3D points by detecting outliers. The method is evaluated on a very controlled setting where both training and testing images are extracted from the same scene including a robotic arm and other objects.

Technical Details.

  • It is clear that COCO was used to compute results in Table 2. However, what datasets were used to obtain the numbers in Tables 3, 4 and 5? If a new experimental setting was really used here as authors claim they did both in their response and in the paper itself, then why are these numbers still identical to the ones in the previous version? Has the experimental setting really changed, with novel training and testing data, as they claim or did the authors simply make this claim to hide the limitations of their actual experimental setting?
  • The claim made by authors that the method is generic and better than some existing methods was not supported by quantitate results on more generic data. The revised version only provides new qualitative results for their method on a more open scene.
  • The paper only shows qualitative results for the proposed method, but never makes a qualitative comparison between the proposed method and existing methods.
  • Reconstruction Figures. Overall, it is never explained how are these visualizations generated. How are the estimated 3D points per pixel being mapped to form a RGB image? The position of a point in 3D space has no direct relation with the original color /texture of the image. One can understand that the raw NN 3D point estimation is worse than the refined estimation. However, it is unclear how the raw output generates these “blurry” reconstructions while the refined one produces the cleaner images.
  • It’s the term W_D missing in equation 3?
  • What is the matrix A used in equation 7? The matrix P is defined right away, but the term A is never defined and it is used to compute equation 8 as well.
  • Section 2.4. P is defined as a “3D reconstruction point”, but later it seems to actually represent a SET of points {p_1, … p_i, … p_m}.
  • Section 2.4. The whole explanation is a lot clearer now, but a couple of details can still be added for further clarification. For example, before providing any math, it might be good to describe the whole procedure at a high level. Currently, this is partially done but after all math has been provided. The goal of the procedure is simply to estimate the density around each point. The cut-off distance defines a “Neighborhood” for each point which can be used to decide which points should end up in the same cluster. After establishing the neighborhoods, the method sets up some sort of hierarchy between them as defined in equation 11. Each point Is assigned a higher value this the hierarchy if the point with the next higher density is far away. Points at the top of this hierarchy become cluster centers. Points at the bottom of the density-based hierarchy can be considered outliers and can be removed. Currently, the readers need to figure out some of these ideas, but providing some context here might be really helpful for them.

Format, Presentation, and Language

  • Algorithm 1. It still does not quite follow a standard / clear pseudo-code notation. The following changes can help:
    • Line 5. It will clearer if a for loop style notation is used here. For example, it could say “for each p_i” (next line)(TAB)”use formula (10) To calculate \rho_i”. Note that “x_i” is used in the original algorithm but that seems to be an error, it should be p_i.
    • Also, formula 11 requires formula 10 to be computed first for all x_i, so a separated “for” loop might be a better representation of this. Current line 6 and 7 actually would fit well within the scope of the for loop for formula 11.
    • Line 13. Same as previous comments. “For each p_i \in P_{in}” (next line) “[TAB] use formulas ….”. Line 14 should be within this loop, but “return \hat{P} should be the last line of the algorithm.
  • Some Tables are split into two pages, with only one row on second page. This should be avoided on future versions.
  • Some language related things to check:
    • depended on the approximation of M.”
    • “The YOLACT++ network is used to segment the visual information to extract the visual information that needs to be reconstructed.”
    • “Figure 1 and Figure 2 in the following chapters.

Author Response

(The authors gave the same response as above.)
